

# miR-27b attenuates apoptosis induced by transmissible gastroenteritis virus (TGEV) infection via targeting runt-related transcription factor 1 (RUNX1)

Xiaomin Zhao[*], Xiangjun Song[*], Xiaoyuan Bai, Naijiao Fei, Yong Huang, Zhimin Zhao, Qian Du, Hongling Zhang, Liang Zhang and Dewen Tong

College of Veterinary Medicine, Northwest A&F University, Yangling, Shaanxi, China
[*] These authors contributed equally to this work.

## ABSTRACT

Transmissible gastroenteritis virus (TGEV), belonging to the coronaviridae family, is the key cause of the fatal diarrhea of piglets and results in many pathological processes. microRNAs (miRNAs) play a key role in the regulation of virus-induced apoptosis. During the process of apoptosis induced by TGEV infection in PK-15 cells, the miR-27b is notably down-regulated. Thus, we speculate that miR-27b is involved in regulating the process of apoptosis in PK-15 cells. In this study we demonstrated that the over-expression of miR-27b led to the inhibition of TGEV-induced apoptosis, reduction of Bax protein level, and decrease of caspase-3 and −9 activities. Conversely, silencing of miR-27b by miR-27b inhibitors enhanced apoptosis via up-regulating Bax expression and promoting the activities of caspase-3 and −9 in TGEV-infected cells. Subsequently, the runt-related transcription factor 1 (RUNX1) is a candidate target of miR-27b predicted by bioinformatics search. We further identified that the miR-27b directly bound to the 3′ UTR of RUNX1 mRNA and suppressed RUNX1 expression, which indicates RUNX1 is the direct target gene of miR-27b. The over-expression of RUNX1 increased apoptosis and knockdown RUNX1blocked apoptosis of viral-infected cells via regulating Bax expression and the activities of caspase-3 and −9. Our data reveal that miR-27b may repress the mitochondrial pathway of apoptosis by targeting RUNX1, indicating that TGEV may induce apoptosis via down-regulating miR-27b and that miR-27b may act as a target for therapeutic intervention.

## INTRODUCTION

TGEV, a member of *Coronaviridae* family, is an enveloped virus with a positive-sense single-stranded RNA genome (*Weiss & Leibowitz, 2011*). TGEV infection primarily causes transmissible gastroenteritis (TGE) that is characterized by highly contagious and fatal gastroenteritis for pigs of all ages, especially for piglets under 2 weeks old (*Chae et al., 2000*; *Kim & Chae, 2001*).

Apoptosis is a process of self-destruction in response to a variety of stimuli such as viral infection. Infection of coronavirus such as porcine epidemic diarrhea virus (PEDV)

Corresponding author
Dewen Tong, dwtong@nwsuaf.edu.cn

(*Kim & Lee, 2014*), infectious bronchitis virus (IBV) (*Li, Tam & Liu, 2007*), severe acute respiratory syndrome coronavirus (SARS-CoV) (*Krahling et al., 2009*), may result in host cell apoptosis. We have reported that TGEV infection induced apoptosis via mitochondria mediated apoptotic pathway in PK-15 cells (*Ding et al., 2012*; *Ding et al., 2013*).

miRNAs are a class of small non-coding RNA molecules and may regulate gene expression at post-transcription level (*Bartel, 2009*). Numerous experimental studies have demonstrated that miRNAs play important roles in cell apoptosis (*Wilson & Doudna, 2013*). miR-29b effectively inhibits apoptosis via directly targeting caspase-7 and nuclear apoptosis inducing factor 1 (NAIF1) in Madin-Darby Bovine Kidney (MDBK) cells infected with bovine viral diarrhea virus (BVDV) (*Fu et al., 2014*). miR-27b promotes doxorubicin-induced apoptosis in hepatoblastoma cell line (HepG2) (*Mu et al., 2015*). Moreover, miR-27b is an endogenous inhibitory factor of apoptotic peptidase activating factor 1 (Apaf-1) expression and decreases the apoptotic rate of neurons(*Chen et al., 2014*). miRNAs are also involved in regulating the process of virus-induced apoptosis (*Smith et al., 2012*; *Zhang et al., 2013*). Hs_154 promotes WNV-mediated apoptosis via inhibiting the expression of CCCTC-binding factor (CTCF) and the epidermal growth factor receptor (EGFR) (*Smith et al., 2012*). Our previous studies showed that miR-27b was significantly down-regulated during TGEV-induced apoptosis (*Song et al., 2015*). Thus, we supposed that miR-27b may play a role in apoptosis induced by TGEV infection. In the present study, we investigated the role of miR-27b in TGEV-induced apoptosis in PK-15 cells and demonstrated that the miR-27b attenuated TGEV-induced apoptosis by targeting RUNX1, suggesting TGEV may use miR-27b to regulate apoptosis in PK-15 cells.

## METHODS

### Antibodies, cells and virus

Monoclonal RUNX1 (MAB23991) was purchased from R& D Systems (R& D Systems, Minneapolis, MN, USA). Monoclonal antibodies against Bax (sc-23959) and $\beta$-actin (sc-69879) were purchased from Santa Cruz Biotechnology (Santa Cruz, Inc., Santa Cruz, CA, USA). Horseradish peroxidase (HRP)-conjugated secondary antibody was purchased from Pierce (Pierce, Rockford, IL, US). PK-15 cells were obtained from ATCC (CCL-33) and cultured in Dulbecco's Minimal Essential Medium (DMEM) supplemented with 10% fetal bovine serum (Gibco BRL, Gaithersburg, MD, USA), 100 IU of penicillin, and 100 mg of streptomycin per ml, at 37 °C in a 5% $CO_2$ atmosphere incubator. The TGEV Shaanxi strain was isolated from TGEV-infected piglets by *Ding et al. (2011)*.

### miRNAs quantification by real-time PCR

Total RNA was extracted using Trizol reagent (Invitrogen, Carlsbad, CA, USA) from PK-15 cells Reverse transcription reactions and real-time PCR were performed as described previously (*Song et al., 2015*). The relative quantification of miRNAs was normalized to U6 using the two-ddCt method (*Livak & Schmittgen, 2001*).

## Flow cytometry analysis

Annexin V-FITC Apoptosis Kit (Invitrogen, Carlsbad, CA, USA) was used to detect apoptosis according to the manufacturer's protocol. Briefly, cells were washed twice with ice-cold PBS and resuspended in 500 µL 1×Annexin V binding buffer followed by adding 5 µL Annexin V-FITC and 5 µL PI. After incubation for 30 min at room temperature, analysis was done by flow cytometry (Beckman Coulter, CA, USA).

## Caspase activity assay

Caspases activities were measured by colorimetric assay kits (Keygen Biotech, Nanjing, China) following the manufacture's protocol. Briefly, protein concentrations were measured by BCA Protein Assay Reagent (Vazyme Biotech Nanjing, China). Then 200 mg protein of each sample was incubated with each caspase substrate at 37 °C in a microplate for 4 h. Absorbance at the wavelength of 405 nm was read in microplate spectrophotometer (Infinite 200 PRO NanoQuant; Tecan, Switzerland).

## Luciferase reporter experiments

The 3′ UTR sequence of RUNX1 mRNA containing the target sites of miR-27b was amplified by PCR using the primers of Table S1. The 3′ UTR of RUNX1 mRNA was cloned into psiCHECK-2 (Promega, Madison, WI, USA) between *Xho* I and *Not* I cloning sites to obtain the wild type plasmid RUNX1-WT. The binding sequences of miR-27b seed region in 3′ UTR of RUNX1-WT were mutated following a mutagenesis protocol (*Heckman & Pease, 2007*) to generate RUNX1-mut. miR-27b mimics, miRNA mimics control, miR-27b inhibitors and miRNA inhibitors control were synthesized by Ribo Biotech (RiboBio, Guangzhou, China). miR-27b inhibitors were modified with 2′-O-methyl. The sequences of miRNA in this study were shown in Table S2. For the luciferase reporter assay, PK-15 cells were grown in 24-well plates and then co-transfected with plasmid RUNX1-WT (or RUNX1-mut) and miR-27b mimics (or miR-27b inhibitors) using Lipofectamine 3000 (Invitrogen, Carlsbad, CA, USA). The luciferase activities were detected at 48 h post transfection (hpt) using a Dual-Glo® Luciferase Assay System (Promega, Madison, WI, USA) following the manufacturer's manual.

## RNA interference

Three siRNAs (siRUNX1-1, siRUNX1-2, siRUNX1-3) of RUNX1 were synthesized by GenePharma (GenePharma, Shanghai, China). The most effective siRNA (si-RUNX1-2) was applied for the further experiments. PK-15 cells were transfected with 100 nM RUNX1-specific siRUNX1-2 (Table S2) or irrelevant siRNA using Lipofectamine 3000 (Invitrogen, Carlsbad, CA, US).

## Cell viability assay

Cell viability was detected using Cell Counting Kit-8 (CCK-8) (Vazyme Biotech Nanjing, China). Briefly, cells were seeded in 96-well culture plate and cultured at 37 °C in a humidified atmosphere with 5% $CO_2$. Twelve hours later, cells were transfected with 100 nM siRUNX1-2 or irrelevant siRNAs. The cell viability was tested following the manufacturer's manual at 48 hpt.

## Vector construction

To construct the RUNX1 expression plasmid, the full-length RUNX1 gene was amplified by PCR and was cloned into plasmid pCI-neo (Promega, Madison, WI, USA) between *EcoR I* and *Not I* cloning sites to gain pCI-neo-RUNX1. The primer sequences of PCR are shown in Table S1. PK-15 cells were transfected with pCI-neo (control) or pCI-neo-RUNX1. Cells were infected with TGEV at multiplicity of infection (MOI) of 1 at 48 h post infection (hpi).

## Quantification of mRNAs by real-time PCR

The total RNA was extracted using Trizol reagent (Invitrogen, Carlsbad, CA, USA) according to manufacturer's instructions. A total of 2 μg of RNA was transcribed into cDNA using the First-strand cDNA synthesis kit (Invitrogen, Carlsbad, CA, US). Real-time PCR was carried out using the AccuPower® 2× Greenstar qPCR Master mix (Bioneer, Daejeon, Korea) on Bio-Rad iQ5 Real-Time PCR System. The relative fold changes were calculated using the 2-ddCt method. The real-time PCR primers are shown in Table S3.

## Western blot analysis

Cells were treated with Radioimmunoprecipitation Assay (RIPA) lysis buffer containing phenylmethyl sulfonylfluoride (PMSF). Protein concentration was tested with BCA Protein Assay Reagent (Pierce, Rockford, IL, USA). The proteins were separated on a 12% sodium dodecyl sulfate-polyacrylamide (SDS-PAGE) gel and subsequently transferred to a polyvinylidene difluoride (PVDF) membrane (Millipore Corp, Atlanta, GA, USA). The membrane was blocked with 5% non-fat dry milk for 2 h at room temperature and then incubated with the primary antibodies overnight at 4 °C. The HRP-conjugated secondary antibodies were used to incubate for 1 h at room temperature. The blotting was visualized using Enhanced Chemiluminescence (ECL) reagent.

## Statistics

All data are representative of results from at least 3 separate experiments. All data points are the average of triplicates, with error bars representing standard deviations. The difference between two groups was statistically analyzed by a two-tailed Student's $t$ test using SPSS software. $P$ value <0.05 was considered significant.

# RESULTS

## miR-27b attenuates TGEV-induced apoptosis

To investigate whether miR-27b is involved in the apoptosis induced by TGEV, the cells were transfected with miR-27b mimics or miR-27b inhibitors and infected with TGEV at a MOI of 1 at 24 hpt. The apoptotic rate was analyzed by flow cytometry at 24 and 48 hpi. The results showed that the over-expression of miR-27b led to a decrease of apoptotic rate and the down-expression of miR-27b increased apoptotic rate at 24 and 48 hpi (Fig. 1A), indicating that miR-27b attenuated apoptosis induced by TGEV infection in PK-15 cells. We previously found that TGEV infection could cause PK-15 cell apoptosis through mitochondria-mediated pathway and FasL-mediated apoptotic pathway

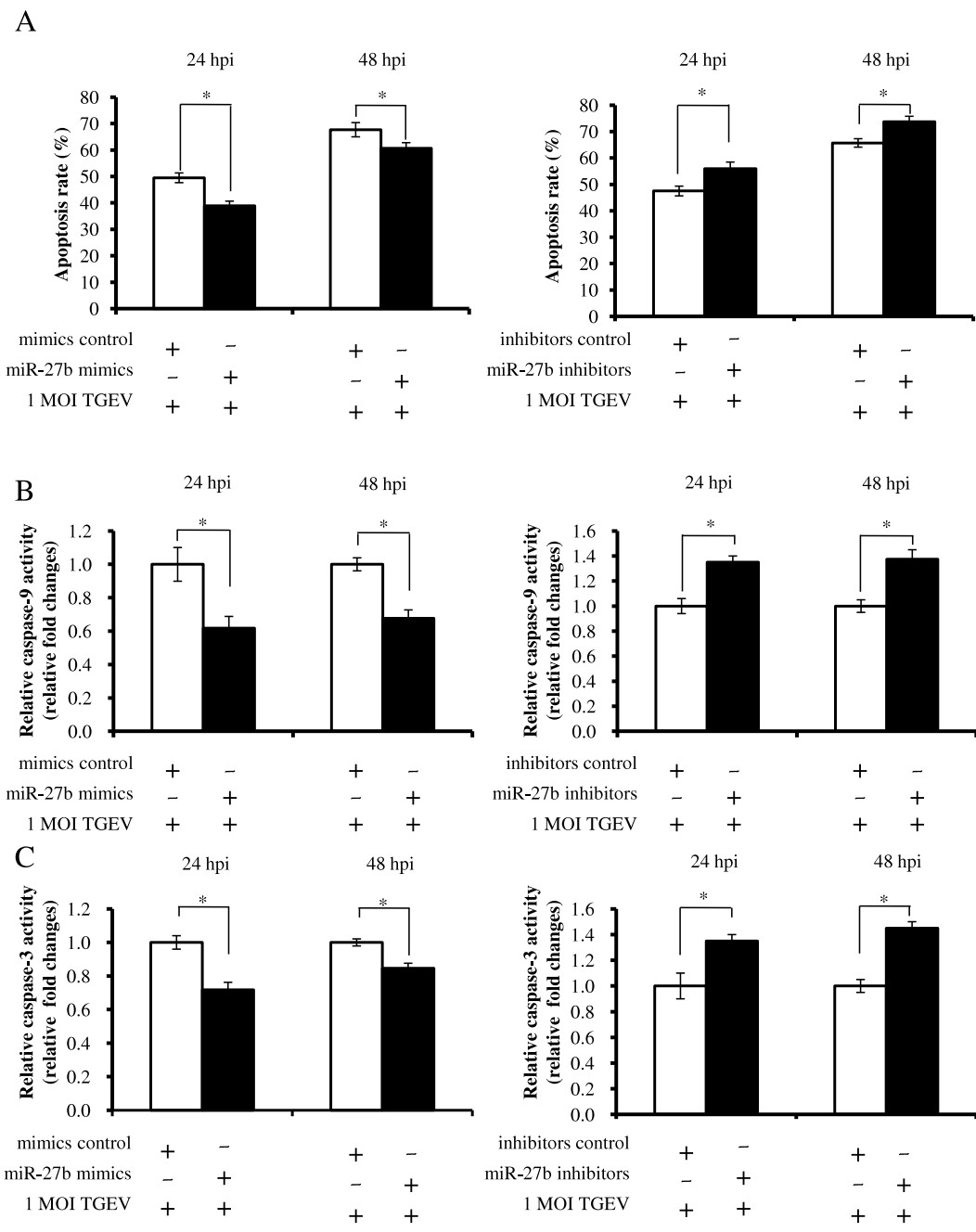

**Figure 1 The effect of miR-27b on apoptosis induced by TGEV.** (A) Effect of the miR-27b mimics and inhibitors on apoptosis induced by TGEV. The apoptosis of PK-15 was analyzed via flow cytometry at 24 and 48 hpi. (B) Detection of caspase-9 activities in PK-15 cells at 24 and 48 hpi. (C) Detection of caspase-3 activities in cells at 24 and 48 hpi. *$P < 0.05$ in comparison with the control. **$P < 0.01$ in comparison with the control.

(*Ding et al., 2012*). Therefore, we analyzed caspase-9 activity of mitochondria-mediated pathway and caspase-8 activity of FasL-mediated apoptotic pathway. However, miR-27b did not affect the caspase-8 activity during TGEV infection (data not shown). As expected, the caspase-9 activity was decreased by miR-27b mimics and up-regulated by miR-27b
inhibitors in the TGEV-infected PK-15 cells (Fig. 1B). Moreover, we investigated the effect of miR-27b on caspases-3 activity in TGEV-infected PK-15 cells. The caspase-3 activity was decreased by miR-27b mimics and increased by miR-27b inhibitors in the TGEV-infected cells compared with control (Fig. 1C). Taken together, these results suggest that miR-27b plays a negative role in the TGEV-induced apoptosis in PK-15 cells via mitochondria-mediated pathway.

### RUNX1 is the direct target of miR-27b

To identify the targets that affect TGEV-induced apoptosis, we predicted the targets of miR-27b using TargetScan and miRanda and found that RUNX1 was a potential target of miR-27b (*Song et al., 2015*). Moreover, RUNX1 was reported to regulate apoptosis (*Wu et al., 2012*). Thus, we investigated the direct interaction between miR-27b and 3′ UTR of RUNX1 mRNA through dual-luciferase assay. The sequence of the 3′ UTR of RUNX1 was obtained by PCR and the two binding sites of miR-27b seed region in the 3′ UTR of RUNX1 were mutated with a 3-bp substitution. The 3′ UTR of RUNX1 and its mutational version were respectively sub-cloned into the 3′ UTR of the *Renilla* luciferase dual-luciferase plasmid psiCHECK-2 (Figs. 2A and 2B). The constructs were co-transfected into PK-15 cells with miR-27b mimics (or negative control) or miR-27b inhibitors (or negative control). Compared with the miRNA mimics control, introduction of miR-27b mimics decreased the RUNX1-WT reporter activity (Fig. 2C). To determine whether miR-27b inhibits RUNX1 expression, miR-27b mimics or miRNA mimics control were transfected into PK-15 cells. RUNX1 expression was assessed by western blot. The result showed that miR-27b decreased expression of RUNX1 in PK-15 cells compared with the control (Fig. 2D). Together, our data conclusively demonstrate that RUNX1 is a direct target of miR-27b in PK-15 cells.

### miR-27b attenuates apoptosis though regulating mitochondrial pathway

RUNX1 shows transactivate effect on Bax expression (*Wu et al., 2012*).We previously found that TGEV infection could cause PK-15 cell apoptosis through mitochondria-mediated pathway (*Ding et al., 2012*). Therefore, we proposed a hypothesis that miR-27b could restrain TGEV-induced apoptosis of PK-15 through mitochondria-mediated pathway by decreasing expression of Bax. In order to confirm this hypothesis, we analyzed the effect of miR-27b on Bax. At 24 hpt of miR-27b mimics or miR-27b inhibitors, the cells were infected with TGEV at 1 MOI. The mRNA and protein levels of Bax were analyzed by real-time PCR and western blotting at 24 and 48 hpi. Over-expression of miR-27b resulted in a decrease of Bax at both mRNA and protein levels, down-expression of miR-27b increased the mRNA and protein expression of Bax (Figs. 3A and 3B). Taken together, these results indicate that miR-27b attenuates TGEV-induced apoptosis by suppressing Bax.

### RUNX1 enhances TGEV-induced apoptosis

To investigate the effect of RUNX1 on apoptosis, we silenced and over-expressed RUNX1 respectively using siRUNX1 and pCI-neo-RUNX1. The RUNX1 mRNA and protein levels were remarkably reduced by siRUNX1-2 (Figs. 4A and 4B); however, the cell viability was

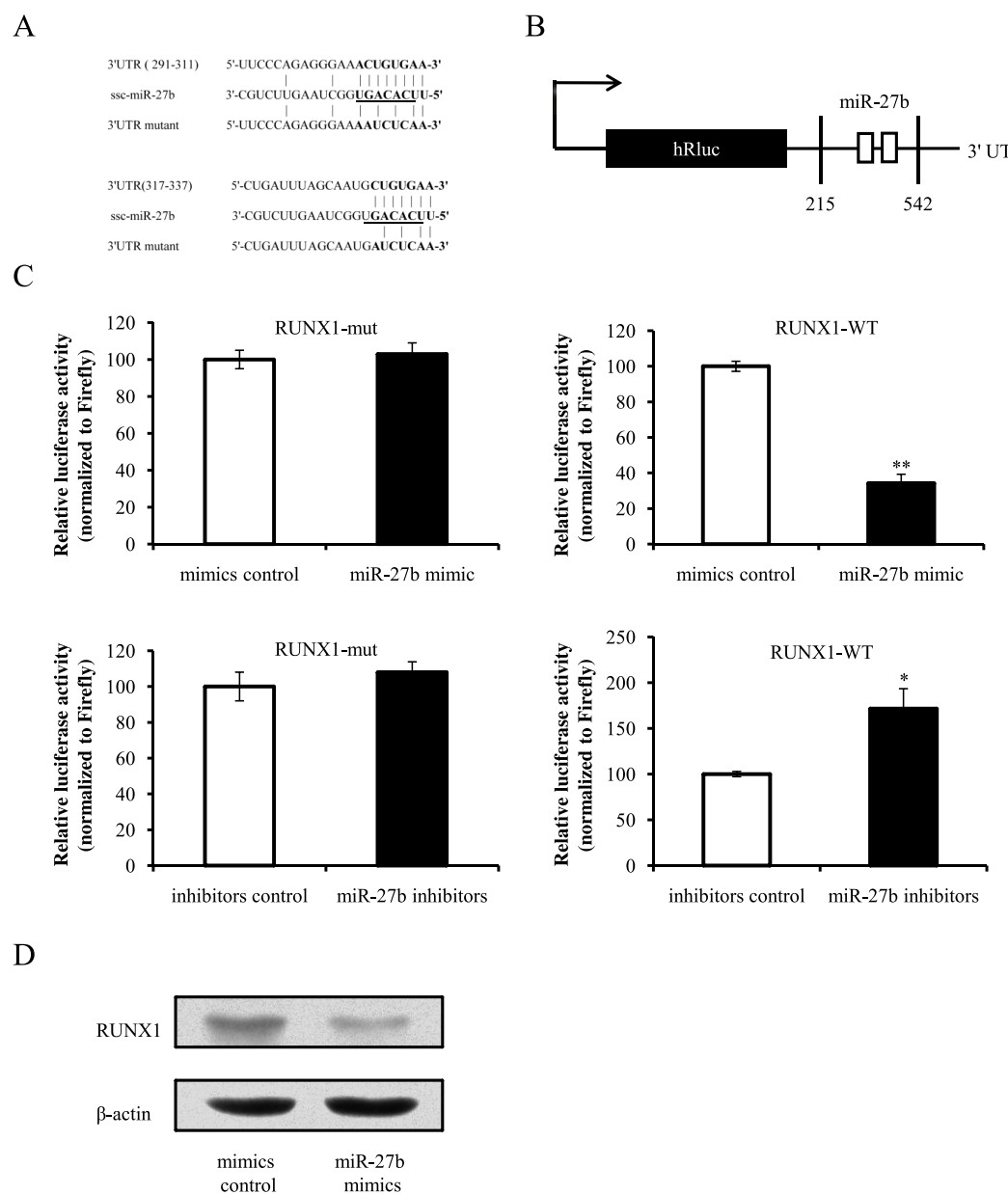

**Figure 2** **miR-27b directly targets the 3′ UTR of RUNX1 mRNA.** (A) Bioinformatic prediction of interaction between miR-27b and the 3′ UTRs of swine RUNX1. For each schematic, the upper sequence is the binding site of miR-27b in 3′ UTRs of swine RUNX1, the middle sequence is the mature miR-27b, and the lower sequence is the mutated sequence of 3′ UTR. The seed sequence is underlined. (B) Schematic drawing of the putative binding sites or mutations of miR-27b binding-sites in 3′ UTR of RUNX1 mRNA. The locations of the potential binding sites or their mutations are presented by blank boxes. (C) The RUNX1 luciferase reporter construct was co-transfected with miR-27b mimics (or negative control) or miR-27b inhibitors (or negative control) into PK-15 cells (normalized to the firefly luciferase activity). Data are expressed as relative luciferase activities to control. (D) Western blot analysis of RUNX1 in cells transfected with miR-27b mimics or mimics control. Data represent means $\pm$ S.D. of three independent experiments. *$P < 0.05$ in comparison with the control. **$P < 0.01$ in comparison with the control.

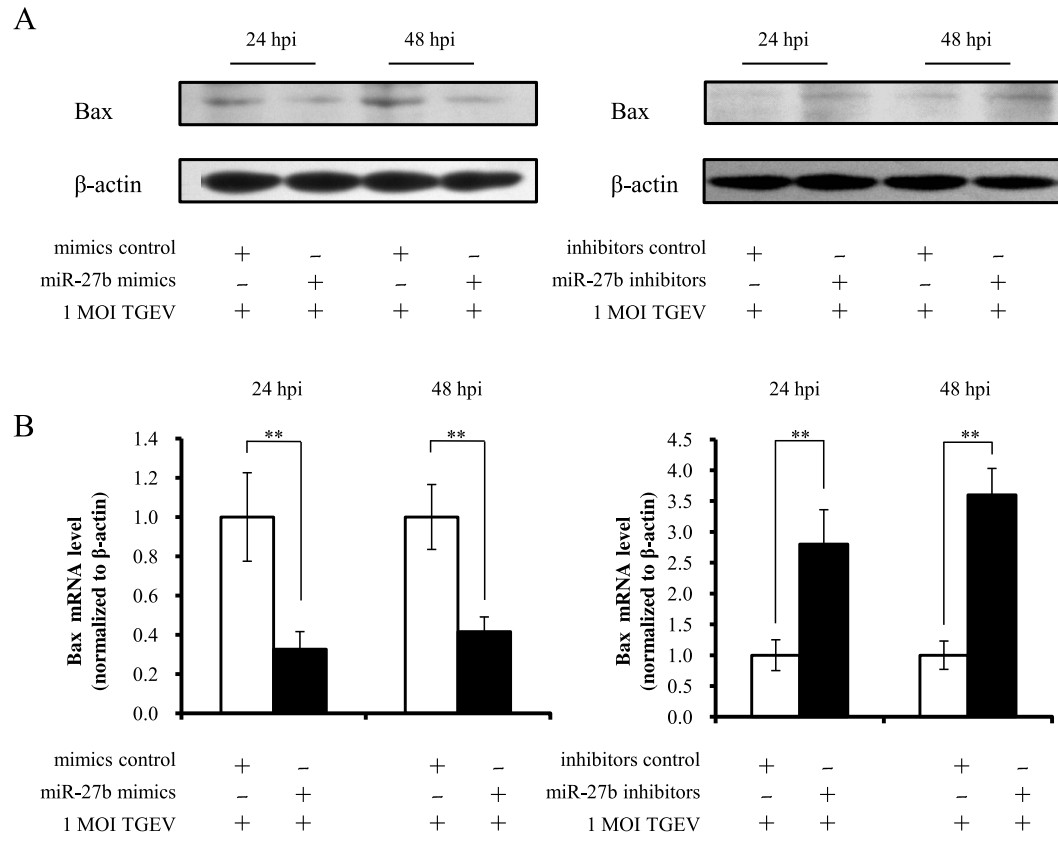

**Figure 3** **miR-27b attenuates apoptosis via mitochondrial pathway.** (A) Western blot analysis of Bax in cells transfected with miR-27b mimics or miR-27b inhibitors. (B) Real-time PCR analysis of the expression of Bax in cells transfected with miR-27b mimics or miR-27b inhibitors. The Bax mRNA level was reduced at 24 and 48 hpi measured by real-PCR (normalized to $\beta$-actin). Data represent mean $\pm$ S.D. of three independent experiments. *$P < 0.05$ in comparison with the control. **$P < 0.01$ in comparison with the control.

not affected in comparison with irrelevant siRNA (Fig. 4C), indicating that the RUNX1 gene was knocked down in PK-15 cells by the siRUNX1-2. In addition, RUNX1 was notably over-expressed using pCI-neo-RUNX1 (Fig. 4D). To determine the effect of RUNX1 on Bax, the RUNX1 were silenced with siRUNX1-2 and over-expressed by pCI-neo-RUNX1 followed by infection with TGEV. The Bax expression was decreased by knockdown of RUNX1 and up-regulated by over-expression of RUNX1 (Fig. 4E). The activities of caspase-9 and -3 were respectively reduced and increased by silence and over-expression of RUNX1 (Figs. 4F and 4G). Therefore, RUNX1 promoted TGEV-induced apoptosis of PK-15 cells. Taken together, our findings suggest that miR-27b shows a negative regulatory effect on TGEV-induced apoptosis of PK-15 cells via targeting RUNX1.

## DISCUSSION

Accumulating evidence has shown that miRNAs play an important role in apoptosis induced by virus infection (*Fu et al., 2014*; *Guan et al., 2012*; *Smith et al., 2012*). We previously found that TGEV infection induced apoptosis in PK-15 cells via mitochondria-mediated apoptosis

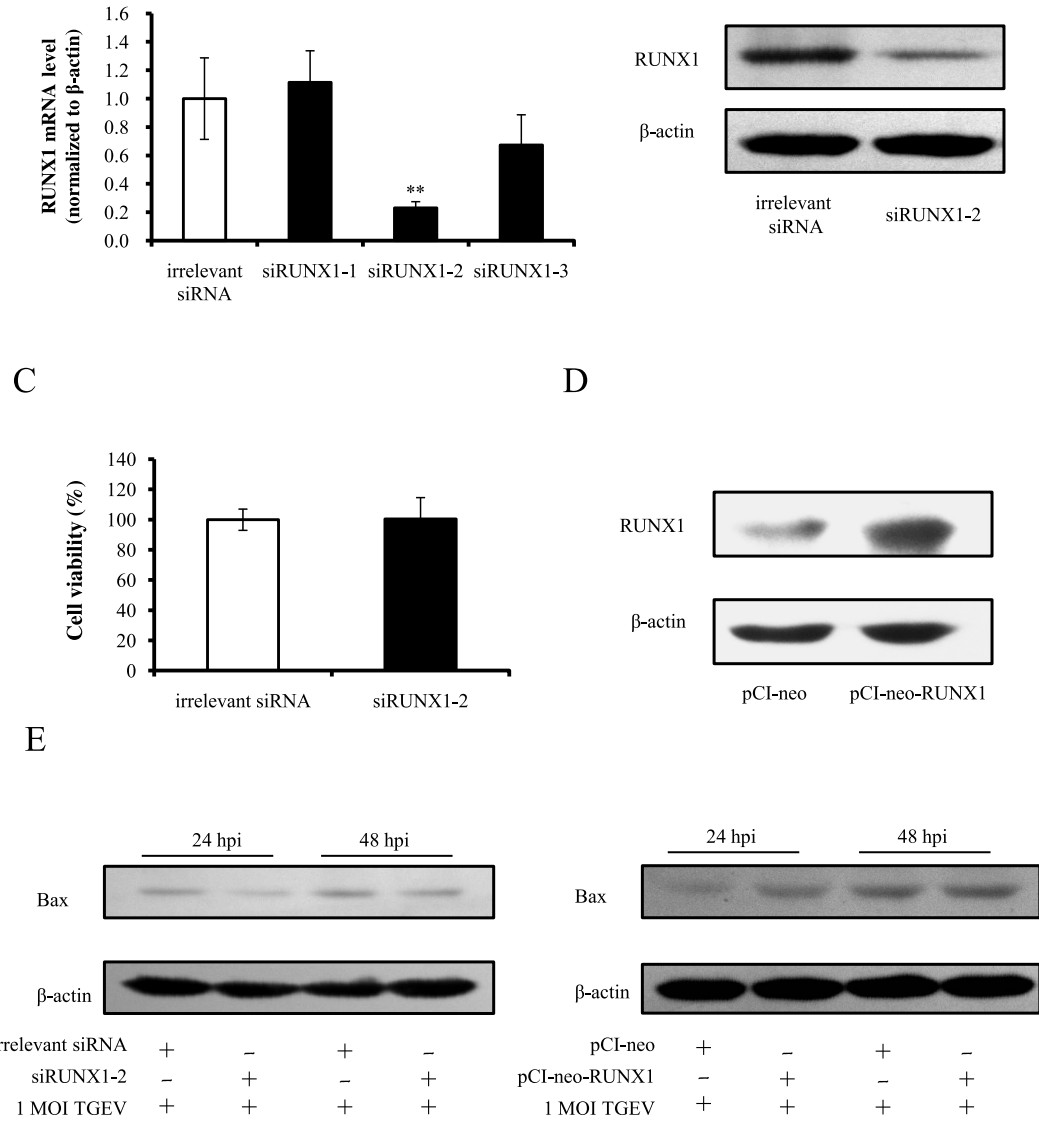

**Figure 4  RUNX1 enhances TGEV-induced apoptosis.** (A) Silencing effect of RUNX1 siRNAs on RUNX1 at mRNA level. PK-15 cells were transfected with RUNX1-specific siRNA or irrelevant siRNA and measured by real-time PCR (normalized to $\beta$-actin). (B) The silencing effect of siRUNX1-2 on RUNX1 expression. (C) The effect of siRUNX1-2 on PK-15-cell viability. The cells were incubated after transfecting with 100 nM siRUNX1-2 for 48 h. Cell viability was evaluated by CCK-8 assay. (D) The over-expression of RUNX1 using pCI-neo-RUNX1. PK-15 cells were transfected with pCI-neo- RUNX1 or pCI-neo vector, and the expression level of RUNX1 was assessed by western blot at 48 hpt. (E) The effect of RUNX1 on the expression of Bax. (F) The effect of RUNX1 on caspase-9 activity. (G) The effect of RUNX1 on caspase-3 activity. Data represent mean ± S.D. of three independent experiments. *$P < 0.05$ in comparison with the control. **$P < 0.01$ in comparison with the control. 

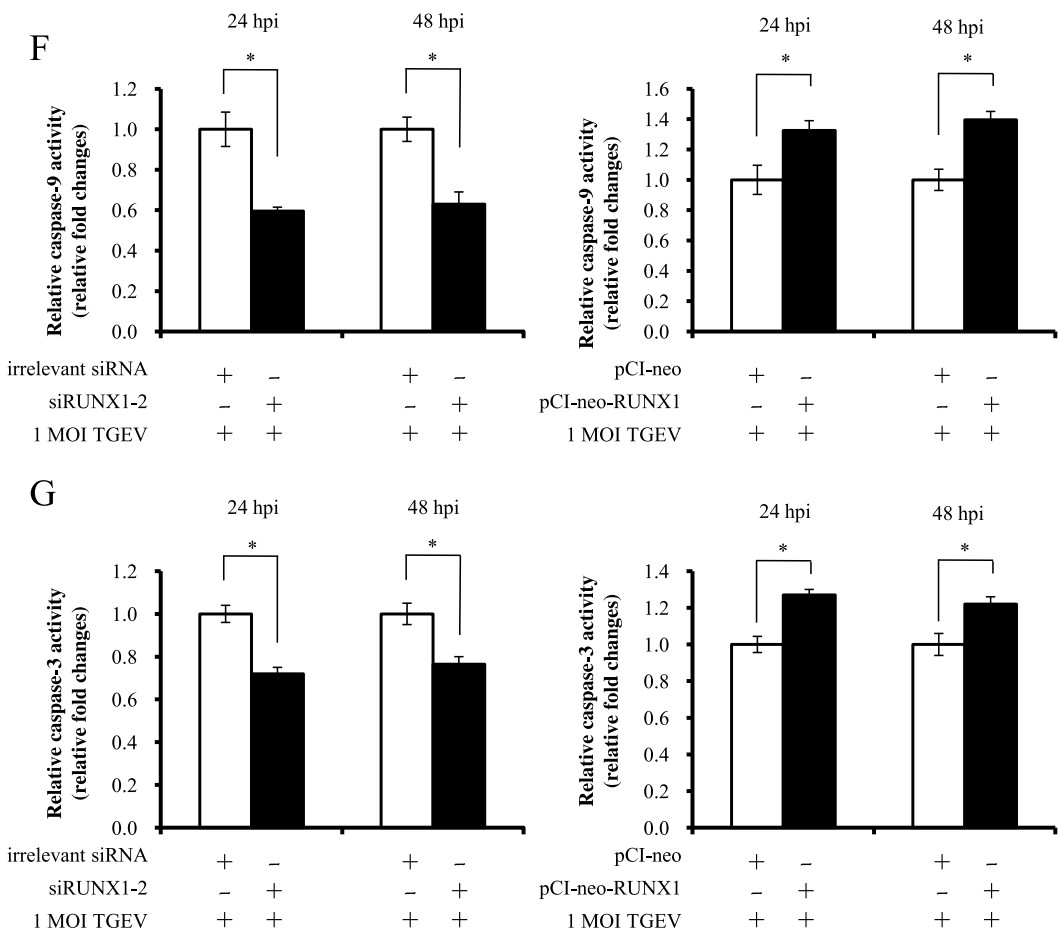

**Figure 4 (…continued)**

pathway (*Ding et al., 2012*). In addition, we reported that miR-27b was significantly down-regulated during apoptosis triggered by TGEV infection in PK-15 cells (*Song et al., 2015*). However, the role of miR-27b in the process of apoptosis induced by TGEV infection is unclear. Here we proved that miR-27b suppressed the TGEV-induced apoptosis via regulating mitochondrial pathway.

During the viral infection, miRNAs may regulate the interaction between virus and host cell via targeting viral or cellular genes. miR-181c, which is down-regulated by HCV infection in hepatocytes, suppresses homeobox A1 expression (HOXA1) and its downstream molecules STAT3 and STAT5 to regulate cell growth by targeting E1 and NS5A sequences of HCV (*Mukherjee et al., 2014*). miR-1236 represses HIV infection by inhibiting transcription of HIV-1 viral protein R-binding protein (VprBP) in monocytes (*Ma et al., 2014*). We assessed the effects of miR-27b on the replication and transcription of TGEV genes and found that the replication and transcription of TGEV structural and non-structural genes were not affected by miR-27b (Data not shown). We tested the effects

of miR-27b on apoptotic rate of TGEV-infected PK-15 cells. The results showed that miR-27b decreased TGEV-induced apoptosis. We previously reported that miR-27b was decreased by TGEV infection (*Song et al., 2015*), so we propose that miR-27b may present an antagonistic effect with TGEV infection on apoptosis via regulating mitochondrial pathway. An analogous situation has been reported that miR-27b alleviate hypoxia-induced neuronal apoptosis(*Chen et al., 2014*). It indicates that miR-27b not only down-regulates cancer cell apoptosis, it also suppresses virus-induced apoptosis.

Apoptosis is primarily modulated by two pathways, extrinsic pathway and intrinsic pathway (mitochondrion-mediated), which regulate cell death involving the continuous activation of caspases (*Lavrik, 2010*). Caspase-8 and caspase-9 are factors of extrinsic pathway and mitochondrial pathway respectively (*Sakamaki et al., 2015*; *Thornberry & Lazebnik, 1998*). Bax, a member of the BCL-2 family, functions as an apoptotic activator to activate caspase-9 and -3 (*Chipuk et al., 2010*; *Ding et al., 2012*). In this study, we showed that miR-27b inhibited Bax expression and reduced the activities of caspase-9 and -3 during TGEV-induced apoptosis. We demonstrated that miR-27b regulated apoptosis via down-regulation the expression of Bax and activation of caspase-9 and caspase-3, implying that miR-27b are involved in regulating TGEV-induced apoptosis via regulating mitochondrial pathway.

The miRNAs suppress gene expression by imperfectly binding to the 3′ UTR of target mRNA in mammalian cells. The imprecise binding make each miRNA has the ability to target multiple mRNAs. For example, miR-27b could suppress adipogenesis in hMADS cells by targeting peroxisome proliferator-activated receptor gamma (PPAR$\gamma$) (*Karbiener et al., 2009*) and promote differentiation of myeloblasts through targeting RUNX1 (*Feng et al., 2009*). In the present study, we confirmed that RUNX1 is the target of miR-27b during TGEV infection, while whether miR-27b regulates apoptosis via RUNX1 is unclear. RUNX1 is a transcription factor, and play important roles in regulating the growth, development, and/or differentiation of multiple lineages of haematopoietic cells (*Feng et al., 2009*). We found over-expression of RUNX1 reduced Bax and decreased the activities of caspase-9 and -3, and silencing RUNX1 up-regulated Bax and increased the activities of caspase-9 and -3. This suggests that miR-27b regulates TGEV-induced apoptosis via targeting RUNX1, which is an important supplement of RUNX1 function.

In summary, our finding suggests miR-27b represses TGEV-induced apoptosis by directly targeting RUNX1, which may have potential for therapeutic strategies.

### Funding

This work was supported by grants from the Natural Science Foundation of China (Grant No. 31372401, 31472167), the China Postdoctoral Science Foundation (Grant No. 2015M570860), Shaanxi Key and Innovate Group for Science and Technology Program (Grant No. 2013KCT-28), and the Scientific Research Program of Northwest A& F University (Grant No. 2014YB012, 2013BSJJ015, Z111021103). The funders had no role

in study design, data collection and analysis, decision to publish, or preparation of the manuscript.

## Grant Disclosures

The following grant information was disclosed by the authors:

Natural Science Foundation of China: 31372401, 31472167.

China Postdoctoral Science Foundation: 2015M570860.

Shaanxi Key and Innovate Group for Science and Technology Program: 2013KCT-28.

Scientific Research Program of Northwest A&F University: 2014YB012, 2013BSJJ015, Z111021103.

## Competing Interests

The authors declare there are no competing interests.

## Author Contributions

- Xiaomin Zhao analyzed the data, contributed reagents/materials/analysis tools, wrote the paper, prepared figures and/or tables, reviewed drafts of the paper.
- Xiangjun Song performed the experiments, analyzed the data, wrote the paper, prepared figures and/or tables, reviewed drafts of the paper.
- Xiaoyuan Bai, Naijiao Fei, Yong Huang, Zhimin Zhao, Qian Du, Hongling Zhang and Liang Zhang reviewed drafts of the paper.
- Dewen Tong conceived and designed the experiments, contributed reagents/materials/analysis tools, reviewed drafts of the paper.

## Data Availability

The research in this article did not generate any raw data.

## Supplemental Information

Supplemental information for this article can be found online at http://dx.doi.org/10.7717/peerj.1635#supplemental-information.

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
