# Peer review of "miR-27b attenuates apoptosis induced by transmissible gastroenteritis virus (TGEV) infection via targeting runt-related transcription factor 1 (RUNX1)"

_PeerJ, doi:10.7717/peerj.1635_

## Round 0.1 · original submission · Minor Revisions

The following points should be considered:
1 (RUNX1) Line 226-227 “We tested the effects of miR-27b on apoptotic rate”. The results showed that miR-27b decreased TGEV-induced apoptosis. I suggest the authors add which cells on apoptosis.
2 The reagents detailed information should be added in the materials and methods, including the city, and cat number.
3 The reference Chen et al. the volume and pages should be added.

·

Basic reporting

No Comments

Experimental design

No Comments

Validity of the findings

No Comments

Additional comments

In this manuscript, the authors report that TGEV infection induces apoptosis of PK-15 cell, and the apoptosis was regulated by miR-27b targeting RUNX1.
These results are very significant for the understanding of TGEV pathogenesis or therapeutic intervention to TGEV infection. But some minor issues need to be addressed in order to make this manuscript more perfect and before it can be published in Peerj.

MINOR POINTS
1. Lane 31-32: in Abstract, the sentence “up-regulating Bax expression and inhibiting the activities of caspase-3 and -9 in TGEV-infected cells.” This conclusion is opposite with the results. According to the results in Lane 205-207, The Bax expression up-regulated, the activities of caspase-3 and -9 should be increased. So If the “inhibiting” should be replaced by "promoting"?
2. Lane 72, the sentence “suggesting TGEV may use miR-27b to induce apoptosis in PK-15 cells” is an inappropriate description. Maybe “induce” be replaced by “regulate” is better.
3. So many primer sequences in the sentences in Methods are disorderly. Putting them into table(s) seems more clear.
4. Lane 124, “)” should be deleted? Please modify it.
5. Lane 152, what kind of software was used for statistical analysis?
6. Lane 169, there should be a preposition before the “mitochondria- mediated pathway”, such as “via” or “through” etc.
7. Lane 225, “found” should be replaced by "found that ".
8. Lane 228, the reference “(Osthus et al. 2005)” is error. Please modify it.
9. In the figure legend of Fig.2A, “The seed sequence is underlined”, but the underline is missing.
10. In the figure legend of Fig.4, “real-PCR” should be replaced by "real-time PCR ".

Reviewer 2 ·

Basic reporting

No Comments

Experimental design

No Comments

Validity of the findings

No Comments

Additional comments

Xiamin Zhao et al., “miR-27b attenuates apoptosis induced by transmissible gastroenteritis virus (TGEV) infection via targeting runt-related transcription factor 1” . The study evidenced that miR 27b repress the mitochondrial pathway of apoptosis by targeting RUNX1, indicating that TGEV may induce apoptosis via downregulating miR-27b. The study seem well designed and interesting.
The following points that the authors should address:
1 (RUNX1) Line 226-227 “We tested the effects of miR-27b on apoptotic rate”. The results showed that miR-27b decreased TGEV-induced apoptosis.
I suggest the authors add which cells on apoptosis.
2 The reagents detailed information should be added in the materials and methods, including the city, and cat number.
3 The reference Chen et al. the volume and pages should be added.

---

## Round 0.2 · accepted · Accept

The authors have improved the manuscript and in my opinion it is now Acceptable